# Consequences of Land Use Changes on Native Forest and Agricultural Areas in Central-Southern Chile during the Last Fifty Years

Alejandro del Pozo [1], Giordano Catenacci-Aguilera [1,2] and Belén Acosta-Gallo [3,*]

1    Centro de Mejoramiento Genético y Fenómica Vegetal, Facultad de Ciencias Agrarias, Universidad de Talca, Talca 3460000, Chile
2    Centro de Innovación y Desarrollo para Ovinos del Secano—Ovisnova, Facultad de Recursos Naturales y Medicina Veterinaria, Universidad Santo Tomás, Talca 346000, Chile
3    Departamento de Biodiversidad, Ecología y Evolución, Facultad de Ciencias Biológicas, Universidad Complutense de Madrid, 28040 Madrid, Spain
*    Correspondence: galloa@bio.ucm.es

**Abstract:** Chile's central-south region has experienced significant land use changes in the past fifty years, affecting native forests, agriculture, and urbanization. This article examines these changes and assesses their impact on native forest cover and agricultural land. Agricultural data for Chile (1980–2020) were obtained from public Chilean institutions (INE and ODEPA). Data on land use changes in central and south Chile (1975–2018), analysed from satellite images, were obtained from indexed papers. Urban area expansion in Chile between 1993 and 2020 was examined using publicly available data from MINVIU, Chile. Additionally, photovoltaic park data was sourced from SEA, Chile. Field crop coverage, primarily in central and southern Chile, decreased from 1,080,000 ha in 1980 to 667,000 ha in 2020, with notable decreases observed in cereal and legume crops. Conversely, the coverage of export-oriented orchards and vineyards increased from 194,947 ha to 492,587 ha. Forest plantations expanded significantly, ranging from 18% per decade in northern central Chile to 246% in the Maule and Biobío regions. This was accompanied by a 12.7–27.0% reduction per 10 years in native forest. Urban areas have experienced significant growth of 91% in the last 27 years, concentrated in the Mediterranean climate region. Solar photovoltaic parks have begun to increasingly replace thorn scrub (Espinal) and agricultural land, mirroring transformations seen in other Mediterranean nations like Spain and Portugal.

**Keywords:** field crops; forest plantation; Mediterranean climate region; orchards; solar parks; urban sprawl

## 1. Introduction

The world population continues to grow and is expected to reach 9 billion by the year 2025. This means that the demand for housing, food, energy, and natural resources will continue, as well as the use and degradation of natural vegetation, unless a more sustainable development strategy is implemented. Land use change has affected 32% of the global land area in the period 1960–2019 [1] as a result of afforestation and agricultural abandonment in northern countries, and deforestation and agricultural expansion in South America, Africa, and Oceania [1,2]. Globally, forest cover has increased, but in many arid and semi-arid ecosystems, land degradation and bare ground have risen between 1982 and 2016 [3]. In addition, urban areas have increased globally [2], which, together with deforestation, agriculture expansion, and forest plantation with exotic trees, are the main drivers of native forest and biodiversity losses.

Mediterranean ecosystems worldwide are recognised for their abundant biodiversity of vascular plants, which contributes to their role as repositories of genetic resources and

providers of ecosystem services, and their tourism potential [4]. However, these ecosystems are currently experiencing significant and continuous land use changes, exacerbated by the effects of climate change, making them highly vulnerable in the face of global environmental shifts [5].

The Mediterranean climate region (MCR) in central Chile has experienced significant pressure on land use change due to population concentration, agricultural expansion, and forestry activities [6–8]. Consequently, these activities have had detrimental effects on the native forest and agricultural land and on the provision of crucial ecosystem services, including carbon sequestration, nutrient cycling, and overall biodiversity loss [9–12]. Over the past fifty years, population growth has been notably concentrated in the Metropolitana and Valparaiso regions, situated in the central part of Chile. Additionally, the country's economy has seen significant improvement, with the gross domestic product (GDP) per capita rising from US$2531 in 1980 to US$16,265 in 2020 [13]. These economic advancements, coupled with policies promoting economic openness and free markets since the 1980s, have brought about substantial transformations in the agriculture and forest sectors which have yet to be thoroughly analysed.

The objective of this article was to examine the significant changes in land use that have occurred in Chile over the past fifty years and assess their impact on native forest cover and agricultural land. In Chile, agricultural land is limited, covering approximately 3.3 million hectares, which accounts for only 4.3% of the country's total land area. Out of this, only 1.2 million hectares are under irrigation, emphasizing the critical importance of preserving agricultural land for the nation.

## 2. Study Area

The central-south of Chile (30–37° S) presents a typical Mediterranean-type climate characterized by annual average precipitation ranging from 100 to 1000 mm and distinct seasonal variations. The natural vegetation is characterised by sclerophyllous forest in the coastal mountain range (CMR) and foothill of the Andes, thorn scrub (*Espinal*) mainly in the eastern part of the CMR, and Nothofagus forests in the most humid areas of the CMR and the Andes [14] (Figure 1).

*History of Interventions*

Following the Spanish colonization in the 15th century, part of the sclerophyllous forest vegetation underwent a gradual conversion into thorn scrub, known as *Espinal*, dominated by the legume tree *Vachelia caven* (synonym *Acacia caven*) [9]. This transformation resulted from the clearing and fragmentation of the sclerophyllous forest, leading to the establishment of an agroforestry system, the *Espinal*, characterized by various exotic and native herbaceous species [9,15,16]. Additional woody species from the sclerophyllous shrubland, such as *Maitenus boaria* Mol., *Quillaja saponaria* Mol., *Schinus polygamus* (Cav.) Cabr., and others, can still be found, albeit in small, scattered patches across the region.

The impact of human activities on the sclerophyllous forest and *Espinal* have been the result of "waves" of intervention; the first after the arrival of Spaniards in the mid-16th century, and another following the arrivals of other Europeans in the 19th century, which led to a major transformation in the landscape with the development of the agriculture [15]. This was further exacerbated by the large production and export of wheat during the gold rush in California and Australia in the 19th century, mainly in hilly soils in the CMR not adequate for cropping, leading to vast soil erosion and degradation.

The central-south part of Chile has experienced significant transformations in land use and the loss of native forests. Estimations of the vegetation cover and land use change in central and south Chile (35°–43°30′ S) indicate that the native forest cover has reduced from 11.3 million ha in 1550 to 5.8 million ha (51% of the original area) in 2017 [17]. Over the past fifty years, the transformation in land use has been very intense (Figure 1), resulting in substantial reductions in the extent of sclerophyllous forests and *Espinal* areas [18,19]. Indeed, a second wave resulted from a government subsidy for tree planting in 1974, which

promoted the plantation of exotic species like *Pinus radiata* and *Eucalyptus* spp. (Figure 1). Additionally, in recent years, there has been a notable rise in the establishment and expansion of solar photovoltaic parks for energy generation, not only in the northern desert but also in the central-southern region, stretching from Valparaiso to Biobio (Figure 1).

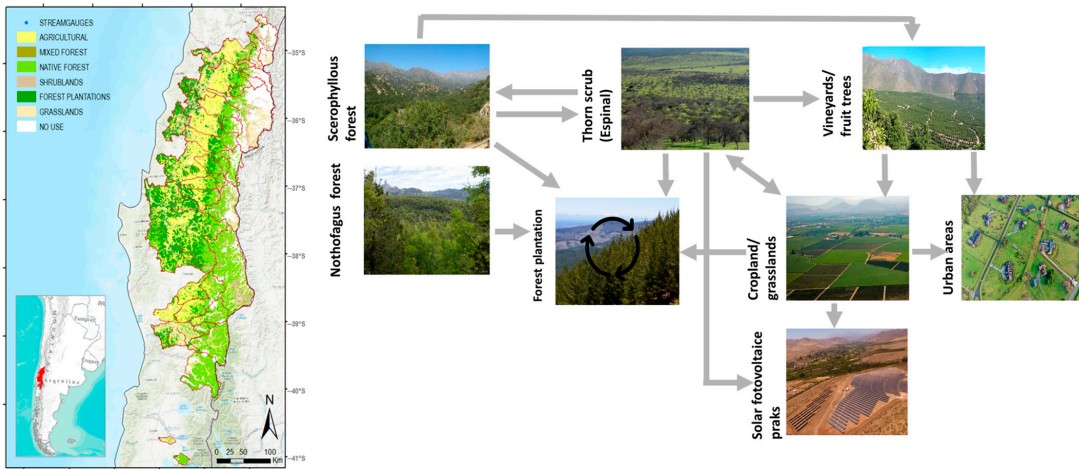

**Figure 1.** Land use changes have occurred in the central-south region of Chile, impacting the Mediterranean sclerophyllous and Nothofagus forests. These forests have been deforested or burned, giving way to a thorn scrub agroecosystem known as Espinal. Additionally, significant areas of native forests and Espinal have been transformed into forest plantations with non-native species, vineyards, or fruit orchards over the past fifty years. The expansion of urban areas has extended to croplands, grasslands, vineyards, and orchards. Furthermore, the emergence of solar parks poses a new threat to agricultural land in the central-south region of Chile. The vegetation map of central Chile was adapted based on the original figure published by Sustainability, MDPI [20].

In addition, the unsustainable utilization of the *Espinal* for agriculture and charcoal production has resulted in a decline in woody cover, specifically *V. caven*, which has led to the exposure of extensive hilly areas to soil erosion, the loss of soil carbon, and a decrease in the diversity and productivity of herbaceous vegetation [9] (Figure 2).

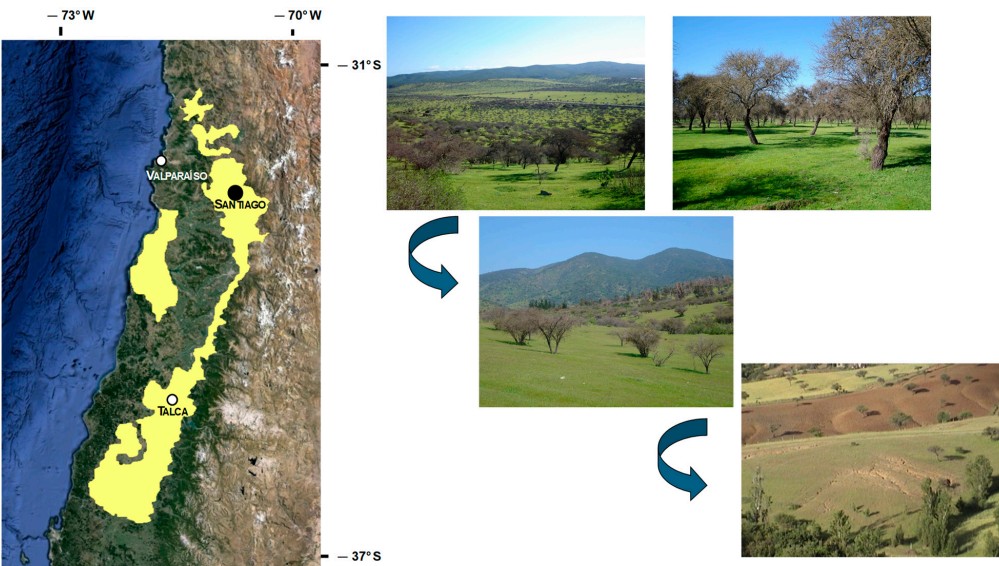

**Figure 2.** Distribution of the *Espinal* agroecosystem (highlighted in yellow) and changes in tree cover, specifically *V. caven*, resulting from activities such as tree-cutting for charcoal production, land clearance for crops, and overgrazing.

## 3. Materials and Methods

Population data for Chile from 1980 to 2020, as well as information on administrative regions in 2020, were acquired from the National Institute of Statistics (INE), Chile [21]. Additionally, data series encompassing field crops, orchards, vineyards, and forest plantations spanning the same period were gathered from the Office for Studies and Agrarian Policies (ODEPA), Chile [22].

Land use changes in central and south Chile from 1975 to 2018, analysed using satellite images, were obtained from published information in indexed papers. The land use types comprised native forest, forest plantation, shrubland, and agriculture and grassland. In addition, the percentages of area change (expressed per 10 years) of native forest, forest plantations, shrubland, and agriculture and grassland over periods of 25–30 years after 1975 were also included.

The expansion of urban areas in Chile and of the three main cities located in the Mediterranean region from 1993 to 2020 was analysed using data from the Ministry of Housing and Urban Development (MINVIU), Chile [23]. Finally, the area and distribution of photovoltaic parks in regions of central Chile (between the Valparaíso and Biobío regions) and the proportion of the type of vegetation replaced by photovoltaic parks in the different regions were obtained from the database of the Environmental Assessment Service (SEA), Chile (https://seia.sea.gob.cl/busqueda/buscarProyectoAction.php; accessed on 28 June 2023).

## 4. Results and Discussion

### 4.1. Changes in Population and Agricultural Land

During the last forty years, the population of Chile has increased from 11.47 to 19.3 million, and more than 50% has been allocated to the Metropolitana and Valparaiso regions, both in the central part of the country (Figure 3A,B). The agricultural land in Chile has been reduced, using only 3.23 million ha (4.3%), while grasslands and shrubs cover 30.24 million ha (39.9%). Indeed, field crop coverage has been reduced from 1080 million ha in 1980 to 667,000 ha in 2020, with a large decline observed in cereal and legume crops (Figure 3C). Wheat has experienced the most significant impact, with the average crop area decreasing from 744,000 ha in the 1960s to 350,000 ha in the 2000s [24], which was further reduced to 226,000 ha in 2020. A similar trend in wheat area has been observed in Portugal, with the wheat area diminishing from 800,000 ha in the 1960s to <200,000 ha in 2005–2008 [25].

Large changes have occurred during the last few decades across most of the irrigated land of central-south Chile, where vineyards and orchards have replaced annual crops (cereals, legumes, etc.); only about 55,000 ha of maize, 20,000 ha of rice, and about 79,000 ha of vegetable crops and 50,000 ha of alfalfa remain in irrigated lands. In rainfed areas, the expansion of exotic tree plantations has replaced field crops, and this has also contributed to the strong decline in cropping area in Chile. Although the grain yield of wheat, maize, and other cereals increased almost linearly after 1980 [24], the total production of grains only covers about 50% of the internal demand; therefore, imports of bread wheat and maize totalled 0.942 and 2.3 million t, respectively, in 2021. Also, the significant reduction in the planted area of grain legumes (Figure 3C) implies that most of them need to be imported.

Orchards and vineyards, primarily focused on exports, have seen a significant expansion from 194,947 ha in 1980 to 492,587 ha in 2020, with notable growth observed in vineyards, stone fruits (especially cherries), and olives (Figure 3D). According to the Agricultural Studies and Politics Office (ODEPA) of the Ministry of Agriculture, the top fruit exports in 2021 were cherries, with a value of USD 1.589 billion FOB, and which were primarily sold to China. Table grapes followed, with a value of USD 927 million FOB, mainly shipped to the United States and China. Apples ranked third, with key buyers including Colombia, the United States, the Netherlands, and India. Blueberries were in fourth place, mostly sent to the United States, the Netherlands, and the United Kingdom. These top four fruit categories, along with kiwi and avocado (ranked fifth and sixth), collectively made up 83% of the total value of fresh fruit exports in 2021. Vineyards covered more

than 140,000 ha in 2021, producing 908.8 million L of wine valued at USD 2037.3 million; 448.2 million L of wine with designation of origin was exported, with a value of USD 1503.9 million.

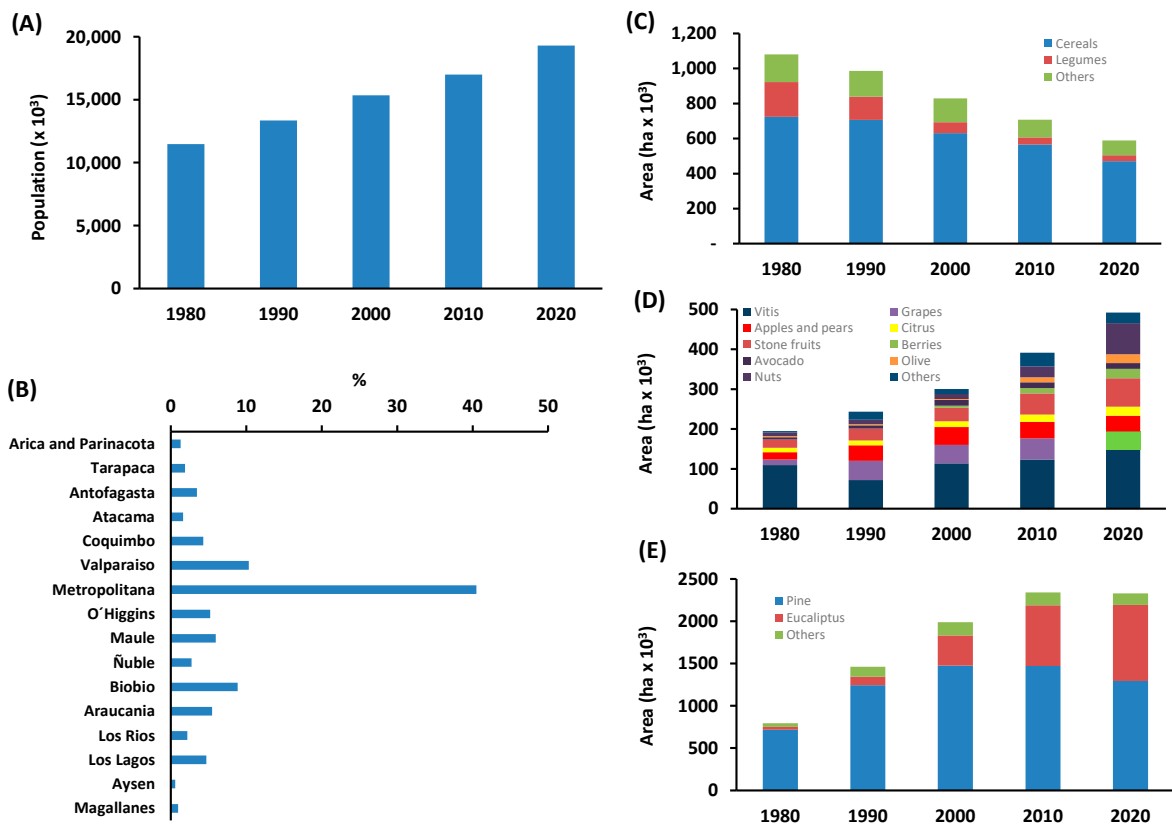

**Figure 3.** Change in Chilean population between 1980 and 2020 (**A**), distribution of the population among administrative regions (Arica and Parinacota is the northernmost region and Magallanes the southernmost) in 2020 (**B**), changes in field crops (**C**), orchards (**D**), and forest plantation (**E**) areas in Chile between 1980 and 2020. Data were obtained from INE (2023) and ODEPA (2023) [21,22].

Despite the necessity to increase the amount of food produced, in Chile and other European countries, there has been a decrease in cropland area in favour of forest and shrubland. According to Jones et al. [25], the proportion of cropland in Portugal has diminished significantly, dropping from a peak of 40% of the total area in the 1960s to 12% by 2006, whereas permanent grasslands have increased. In some regions, there has been also a notable reduction in forest cover over the last two decades, declining from 52% to 22%, primarily attributed to forest fires [25]. Similarly, in Spain, the cropland had a peak of 20.8 million ha in 1970 and then dropped to 17.2 million ha in 2008 [26]. Certainly, Spain stands out as one of the European Union countries profoundly impacted by agricultural land abandonment. This phenomenon has facilitated natural revegetation and afforestation, resulting in a notable expansion of forested areas [27–29]. Also, there has been an intensification in agricultural activity, encompassing the establishment of greenhouses, particularly in coastal areas and major river basins [30].

### 4.2. Changes in Native Forests and Expansion of Exotic Tree Plantation

Studies of land use change in central and south Chile using satellite images from different periods have allowed researchers to evaluate changes in native forests, tree plantations with exotic species, shrubland, and agriculture/grassland, after the 1970s or 1980s (Table 1). Across all of the study sites, there was a high and variable expansion of forest plantations, at a rate between 18% per 10 years in the northern part of central Chile and 246% per 10 years in the CMR of the Maule and Biobío regions (Table 2). This was

accompanied by a 12.7–27.0% reduction per 10 years in native forest (Table 2). According to a study conducted by Nahuelhual et al. [31], which focused on the land cover changes in an area of 529,516 hectares in the coastal range of the Maule and Biobío regions, it was found that tree plantations covered 5.5% of the area in 1975, and this increased to 42.4% in 2007. Furthermore, the study revealed that 41.5% of new plantations between 1975 and 1990, and 22.8% between 1990 and 2007, were established by clearing secondary native forests, indicating that plantation expansion in Chile has directly contributed to deforestation and the decline in biodiversity. Another study in one county (Constitución) of the Maule region, located in the CMR, revealed that between 1955 and 2014, the forest plantation area coverage increased by 1250% (88,649 ha), while native forest coverage was reduced by 23,777 ha, urban areas increased by 2446 ha, and agricultural land decreased by 1992 ha [32]. The shrubland area, including the *Espinal*, was also reduced in these study sites, except in the coastal ranges of the Maule and Ñuble regions (Table 2). More recently, large areas of Mediterranean vegetation have been replaced with high-intensive avocado plantations, on steep slopes with irrigation, severely threatening the sustainability of Mediterranean ecosystems.

Thus, the decline in native forest cover in the MCR can be attributed to the conversion of land for agricultural purposes, the establishment of forest plantations, the expansion of urban areas, and, more recently, the emergence of solar parks for energy production (Figure 1).

Similarly, in southern Chile, the forest plantation expansion ranged between 108% and 379% per 10 years in the Biobio and Araucanía regions, but lower (9.3%) in the Los Rios region (Table 2). The reduction in native forest ranged between 1.9% and 13.4% per 10 years (Table 2). Furthermore, a recent study by Peña-Cortés et al. [33], conducted in two coastal river basins (Budi and Lingue) in southern Chile (Araucanía region), showed that tree plantations increased by 292% and 196% in 1987–2001 and 2001–2015, respectively. In general, the loss of native forests has been due to the expansion of forest plantations towards the Andean and coastal mountain ranges. These studies and others [34,35] clearly indicate that tree plantation has been the main cause of deforestation and biodiversity loss.

The subsidy for tree plantation was for soils of forest aptitude and land without native forest or severely degraded landscapes, and this led to a rapid expansion in forest plantation, increasing from 794,000 ha in 1980 to 2.3 million ha in 2020 (Figure 3E). The last cadastre of native vegetation across the country performed by the National Forestry Corporation (CONAF) indicated that native forests cover 14.73 million ha (19.5% of the land), and forest plantations 3.11 million ha (4.1%) [36]. The latter included all of the forest plantations in the country, including those not oriented to timber production.

### 4.2.1. Impact of Climate Change

Historically, rainfall in this region has exhibited significant year-to-year fluctuations, often associated with the El Niño southern oscillation (ENSO) [37]. Evidence indicates that this Mediterranean climate region has experienced a decrease in precipitation and an increase in temperatures over the past century, likely attributed to climate change [38–40]. This reduction has been further exacerbated by a persistent rainfall deficit since 2010. Central Chile has been experiencing a prolonged period of low precipitation, with annual rainfall shortages ranging from 25 to 45%. While occasional droughts lasting one or two years are common in this Mediterranean environment, the current episode is notable for its duration and widespread impact [40]. These changes in temperature and rainfall patterns are affecting ecosystems, altering species distributions, and leading to habitat loss, as well as affecting the agricultural sector [41].

**Table 1.** Land use changes in central and south Chile, according to different studies using satellite images.

| Study Area | Period | Native Forest (ha) | Forest Plantation (ha) | Shrubland [1] (ha) | Agriculture and Grassland (ha) | Source |
|---|---|---|---|---|---|---|
| Valparaiso, O'Higgins, and Metropolitan regions (central Chile): 1,317,500 ha | 1975–2008 | 1975: 196,107 1999: 130,316 2008: 113,858 | 1975: 106,277 1999: 116,399 2008: 169,537 | 1975: 547,833 1999: 512,407 2008: 428,904 | 1975: 361,848 1999: 237,858 2008: 435,230 | [7] |
| Coastal Range of Maule and Biobio regions (central-south Chile): 578,164 ha | 1975–2000 | 1975: 119,994 1990: 79,643 2000: 39,002 | 1975: 29,579 1990: 96,777 2000: 211,686 | 1975: 193,532 1990: 260,607 2000: 104,151 | 1975: 105,701 1990: 78,482 2000: 124,819 | [42] |
| Coastal Range of Maule and Ñuble regions (central-south Chile): 950,000 ha | 1975–2014 | 1975: 260,338 2014: 122,987 | 1975: 176,850 2014: 356,394 | 1975: 227,123 2014: 359,985 | 1975: 297,145 2014: 122,090 | [43] |
| Biobio and Araucanía regions (south Chile): 2,300,000 ha | 1979–2000 | 1979: 654,069 2000: 469,380 | 1979: 78,581 2000: 705,503 | 1979: 568,143 2000: 314,425 | 1979: 835,447 2000: 631,526 | [6] |
| Coastal Range, Araucaria Region (south Chile): 120,000 ha | 1986–2008 | 1986: 49,445 2008: 36,751 | 1986: 13,299 2008: 45,067 | 1986: 7170 2008: 5614 | 1986: 33,039 2008: 28,764 | [44] |
| Coastal Range of Los Ríos Region (south Chile): 270,000 ha | 1985–2011 | 1985: 188,730 1999: 179,550 2011: 179,280 | 1985: 21,870 1999: 47,425 2011: 58,860 | 1985: 32,130 1999: 19,980 2011: 12,150 | 1985: 25,110 1999: 18,630 2011: 17,550 | [45] |

[1]: mainly Espinal the Mediterranean climate region.

**Table 2.** Estimated percentage of area change (expressed per 10 years) of native forest, forest plantation, shrubland, and agriculture and grassland, for the study period. Data is from Table 1.

| Study Area | Period | Native Forest (%) | Forest Plantation (%) | Shrubland [1] (%) | Agriculture and Grassland (%) |
|---|---|---|---|---|---|
| Valparaiso, O´Higgins, and Metropolitan regions (central Chile): 1,317,500 ha | 1975–2008 | −12.7 | 18.0 | −4.9 | 25.1 |
| Coastal Range of Maule and Biobio regions (central-south Chile): 578,164 ha | 1975–2000 | −27.0 | 246.3 | −18.5 | 7.2 |
| Coastal Range of Maule and Ñuble regions (central-south Chile): 950,000 ha | 1975–2014 | −13.5 | 26.0 | 15.0 | −15.1 |
| Biobio and Araucanía regions (south Chile): 2,300,000 ha | 1979–2000 | −13.4 | 379.9 | −21.3 | −11.6 |
| Coastal Range, Araucaria Region (south Chile): 120,000 ha | 1986–2008 | −11.7 | 108.6 | −9.9 | −5.9 |
| Coastal Range of Los Ríos Region (south Chile): 270,000 ha | 1985–2011 | −1.9 | 9.3 | −23.9 | −11.6 |

[1]: mainly Espinal the Mediterranean climate region.

In the central-southern region of Chile, a combination of rising temperatures, ongoing decreases in precipitation, and land use changes that promote fire-prone landscapes have become the leading cause of anthropogenic fires [46]. Moreover, a recent study highlighted the profound impact of the megadrought in the Mediterranean climate region over the past decade. Specifically, this prolonged drought has adversely affected the growth patterns of two key tree species, *Cryptocarya alba* (Mol.) Looser (Lauraceae) and *Beilschmiedia miersii* (Gay) Kosterm (Lauraceae) of the sclerophyllous forest [47], and has also resulted in a decline in the normalized difference vegetation index (NDVI) and an increase in tree browning [48].

### 4.2.2. Wildfires Incidences

Fire has historically been associated with human activity as a tool to drive landscape transformation [49], but it also represents the consequence of creating conditions that foster the risk of wildfires due to land use change [50]. Thus, the risk of wildfires and the management of this risk become factors that, in turn, influence land use [51].

Fires affect the composition and functioning of ecosystems, thereby influencing the provision of ecosystem services [52,53]. In forests, it has been observed that wildfires and prescribed burns have a negative impact on ecosystem services such as food provisions,

recreation, climate regulation, and water quality, but they can have positive effects on other services such as water supply and soil fertility [54]. The frequency of fires and the time since the last event influence the degree of impact [53]. While fire is a natural factor that shapes many ecosystems [55], human activities often exceed the tolerance for such disturbances, which can harm many species and, consequently, the functioning of ecosystems [56].

The proliferation of *P. radiata* and *Eucalyptus* spp. forest plantations has provided a homogeneous, abundant, and connected reservoir of combustible material that facilitates fire spread [57]. All of these factors converged in the mega-fire of 2017 in Chile [58], which devastated over 480,000 hectares of vegetation, with 54.7% of the affected area being forest plantations compared to the 17.24% of native forests lost in this catastrophe [59]. Despite the severe ecological damage caused by these events, their impact on biodiversity, habitat loss, and fragmentation in the area is relatively lower compared to that caused by land use change associated with forest plantations [60], which also contribute to an increased frequency of fires in the context of climate change [57].

### 4.2.3. Forest Expansion and Impacts in Other Mediterranean and Temperate Regions

In Europe, substantial transformation of forests into agricultural land took place throughout the pre-industrial period, spanning from the Middle Ages until the mid-17th century. More recently, rural depopulation, land abandonment, and the centralization of agricultural activities on fertile soils has set the stage for the regeneration of forests. In addition, a rapid expansion of forest plantations has occurred in the last century, which explains the expansion of forest cover in temperate and Mediterranean European countries [61].

In Spain, 5.6 million ha were planted between 1940 and 2006, initially (1940–1970) with native pine trees, but also using fast-growing species, like *P. radiata* and *E. globulus*, among others. After 1970, the plantation of the native species *Q. robur* and *Q. suber* was performed, but after 1990, there was again an increase in the plantation of fast-growing species such as *E. globulus*, *E. nitens*, *P. radiata*, and *Populus hybrids* [62].

In Portugal, the native forests were dominated by *Quercus* and *Castanea* species (Fagaceae) along with several pine species, which were overexploited, and the country deforested from the 12th century onward, until around the early 20th century [63]. The reforestation process began in the 20th century using mainly *Q. suber* (native) and *E. globulus* [63]. The area of *E. globulus* increased 6.8-fold from the middle 1960s (99,000 ha) to 2005 (740,000 ha); in 2005, *Pinus pinaster* covered 885,000 ha, *E. globulus* covered 740,000 ha, and *Q. suber* covered 716,000 ha [63].

Contrasting with Chile, where the primary approach to forest expansion involves the cultivation of two exotic species, Spain and Portugal have diversified their strategy by establishing plantations with a variety of both native and exotic species.

### 4.3. Expansion of Urban Areas

The expansion of cities can have positive impacts on areas that were previously rural, such as economic development and promoting job creation [64]. However, when this expansion is not regulated, it is referred to as urban sprawl, which is primarily associated with negative impacts on the territory. This concept refers to an uncontrolled process of urban invasion into the peripheral rural zones of cities, associated with increased dependence on cars, the extension of infrastructure, and the fragmentation of agricultural land [65].

Urban sprawl is a common phenomenon in most countries, triggered by many factors, including population growth and migration, land use policies, and housing preferences, among others. It can have detrimental effects, especially in Mediterranean climates, such as threatening the water security of the region and contributing to desertification [66,67]. In recent decades, Spain has experienced an uncontrolled and disproportionate urban sprawl, marked by irreversible soil sealing that has contributed to desertification. The conversion of rural to urban land in Spain has surpassed that of any other EU member state [27]. Urban advancement is often associated with the irreversible loss of fertile soil for agricultural activity [68].

Chile has experienced substantial urban expansion, propelled by multiple factors, including population growth, economic progress, and alterations in land use. In the last 27 years, the urban area has grown by an impressive 91%, primarily concentrated in the Mediterranean region (Figure 4). The Metropolitan region, which encompasses the capital city of Santiago, has had the most extensive expansion, increasing by 34,903 ha from 1993 to 2020 (Figure 4B). Another study showed that the area of the capital Santiago increased 124% between 1997 and 2013 (from 60,130 to 134,750 ha), while agricultural land decreased by 13% during the same period [69]. The area expansion of another seven cities (Antofagasta, Copiapó, La Serena, Concepción, Temuco, Puerto Montt, and Coyhaique) distributed between 23° S and 45° S has ranged between 16.6% (Concepción) and 168.4% (Puerto Montt) over the last 30 years (since 1985–1986) [70].

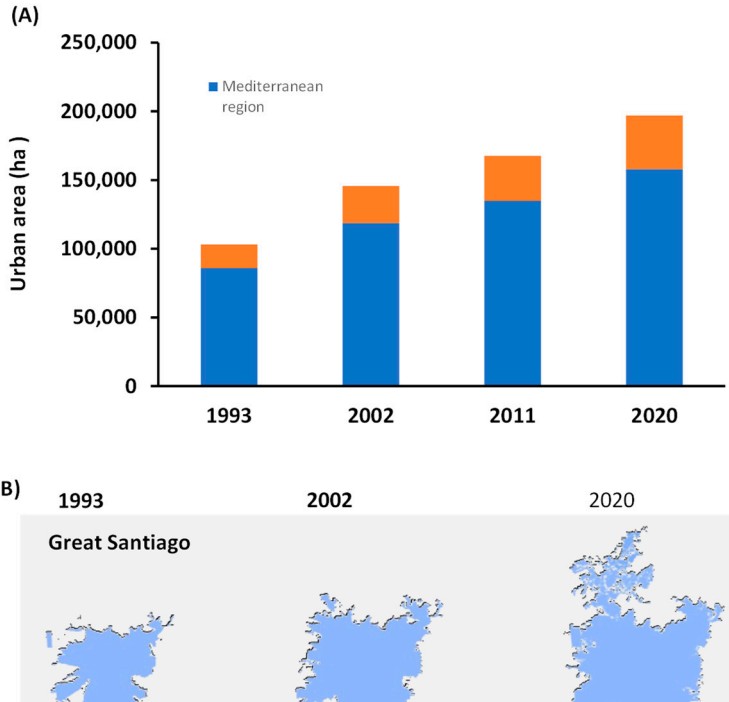

**Figure 4.** (**A**) Expansion of urban areas in Chile and (**B**) the three main cities located in the Mediterranean region from 1993 to 2020. Source: [23].

Another concerning phenomenon transforming agricultural lands in central Chile is the uncontrolled proliferation of "rural residential plots", which are rural properties with a surface area of 5000 m$^2$ primarily used as second homes for the middle and upper classes [71]. According to these authors, there are nearly 140,000 ha of recreational states in the Metropolitan region, and 8000 ha were converted between 2013 and 2018.

The popularity of these properties originates from Decree Law 3516 of 1980, which allows the subdivision of land beyond urban limits while maintaining productive activities

such as agriculture or forestry. However, this latter purpose was never realized, and the sale of land for residential use became the primary result of this instrument [72]. Other factors that promote urban sprawl include weak regulatory policies, the absence of urban regeneration policies, improvements in road infrastructure, and the formation of conurbation areas, among others [73]. This, combined with the voracity of the real estate market, contributes to a situation that promotes segregation, generates environmental problems, and compromises the sustainability of urban food systems [71,74].

Urban sprawl has proven highly lucrative for landowners and construction entrepreneurs. Without effective regulation and the delineation of construction zones, it becomes challenging to impede the transformation of land use and its consequential impacts on native forests and agricultural land.

### 4.4. Solar Photovoltaic Park

A solar photovoltaic park is a large-scale facility designed to generate electricity from sunlight using photovoltaic technology. These parks consist of an array of solar panels, also known as solar modules or solar panels, which convert sunlight into electricity through the photovoltaic effect.

In Chile, the regions comprising northern Chile offer the best climatic conditions for the installation and operation of solar parks due to their elevated levels of radiation and nearly year-round clear skies [75]. Therefore, a large number of photovoltaic parks have been installed in the northern zone of Chile, in a desert environment where they have minimal environmental impact. However, over the last decade, there has been an explosive increase in the number of solar park projects on agricultural land in the central and southern areas of Chile. The main motivation for energy companies is to bring energy generation closer to major consumption centres and thus reduce transmission costs. Also, energy companies are offering farmers lucrative long-term contracts spanning 30 to 40 years, leading to the conversion of agricultural land for solar energy purposes.

According to the database of the Environmental Assessment Service (SEA), in the central zone of Chile (between the Valparaíso and Biobío regions), 244 photovoltaic park projects have been approved since 2015, contributing a total of 3687 MW to the central interconnected system, with an investment exceeding 4000 million dollars. The total area of photovoltaic parks installed and under construction is 6840 ha, with about one-third (2151 ha) of this area being in the Metropolitan region, followed by the O´Higgins and Maule regions (Figure 5A). The average area of solar parks is still low (28.2 ha), and the mean installed potential is 15.2 MW. The largest solar parks cover about 380 ha, but much bigger (>1000 ha) parks are projected in the O´Higgins region. The operational lifespan of the parks is similar in all regions, with an average of 31 years.

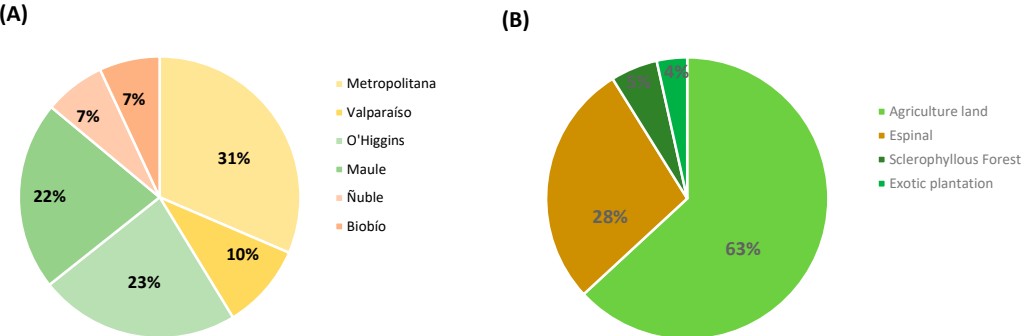

**Figure 5.** (**A**) Distribution of area of photovoltaic parks in regions of Central Chile; and (**B**) proportion of the type of vegetation replaced by photovoltaic parks in different regions of Central Chile.

In terms of land area converted into solar parks, 4317 ha (63%) previously employed for agricultural purposes have been designated for the generation of photovoltaic energy over the last seven years (Figure 5A). Additionally, there has been a conversion of 1916 ha

of the *Espinal* for the same purpose, accompanied by a loss of 367 ha of sclerophyllous forests (Figure 5B). The Metropolitan region, while experiencing a comparatively smaller loss of agricultural land due to solar park conversion, has had the highest proportion of replacement of Espinal and sclerophyllous forests. Conversely, in the Biobío region, nearly all of the currently utilized surface area for photovoltaic parks was previously agricultural land. In the O'Higgins, Maule, and Ñuble regions, although in a smaller proportion compared to other land uses, there have been instances where portions of forest plantations were converted into photovoltaic parks following harvesting.

One of the main drivers for converting agricultural land into surfaces for generating unconventional renewable energy is water scarcity, which decreases energy generation by hydroelectric plants, reducing the energy supply, while also negatively impacting the productive potential of crop fields, making agricultural production riskier, and making the option of leasing land for the location of solar parks more attractive [76]. It is also perceived as an economical solution for many farmers facing drought [76].

The removal of vegetation cover at the solar park sites, coupled with soil management practices in these systems aimed at preventing the growth of herbaceous species, limits the soil's carbon sequestration capacity and promotes the release of accumulated carbon into the atmosphere [77]. In addition, the preparation of the sites for photovoltaic parks and the associated construction practices result in soil disturbance, compaction, reduced water infiltration, and habitat fragmentation for wildlife species, potentially causing serious long-term issues such as desertification, biodiversity loss, and the formation of heat islands [78].

An alternative to using agricultural lands involves employing agrivoltaic systems (AVs). AVs utilize the land for both generating photovoltaic energy through solar panels and conducting pastoral or cropping activities [79–81]. The most basic AVs involve incorporating grazing beneath and between solar panels; sheep or goats are employed to maintain low grasslands, replacing the need for mowing machines. In return, they benefit from forage and shade for resting. Other AVs involve using elevated panels above crops or orchards while adjusting the panel density or tilt. Additionally, some AVs utilize vertically oriented panels with bifacial photovoltaic panels, providing more space for cropping. This vertical panel has the advantage of having a smaller impact on crop growth due to shading and micro-climatic conditions under the panels. As stated by Al Mmun et al. [80] and global research trends in Avs [81], combining farming with energy production can yield favourable economic, social, and environmental benefits while offering practical and viable solutions to the increasing competition for land resources (see Ravilla et al. [82]).

## 5. Conclusions

This study focused on analysing land use changes in Chile over a fifty-year period and evaluating their effects on native forest cover, ecosystem services, and agricultural land. The results indicated that land use change has severely reduced sclerophyllous forest and *Espinal* areas, as well as agricultural land, particularly in the Mediterranean climate region. This has mainly been a consequence of the expansion of forest plantations, urban sprawl, and, lately, photovoltaic parks.

The sclerophyllous forest and the *Espinal* are exceptionally biodiverse and are recognized as a global hotspot for biodiversity conservation. They also function as carbon sinks, although their significance is diminished by factors such as water scarcity and the extent of vegetation cover. Protection of these ecosystems is limited, with only a small percentage under protection, leading to concerns about their preservation and their ability to provide vital ecosystem services. Forest plantations, while reducing timber extraction from native forests and helping with soil erosion control, have significant negative effects on the landscape. Furthermore, the central-southern region of Chile is experiencing the effects of climate change, marked by rising temperatures, ongoing decreases in precipitation, and shifts in land use that promote the development of fire-prone landscapes. These elements have become the predominant factors behind human-induced wildfires in this region.

Urban expansion in Chile, driven by population growth, economic development, and land use changes, has led to significant urban sprawl, particularly concentrated in the Mediterranean region. These expansions, primarily seen in cities like Santiago and seven others, result in a loss of fertile soil, threatening water security, and contributing to desertification, which is particularly problematic in Mediterranean climates. Additionally, the proliferation of "rural residential plots" for second homes exacerbates the issue. Weak regulations, the absence of urban regeneration policies, improved road infrastructure, and conurbation formation further fuel this situation, leading to segregation, environmental challenges, and compromised urban food systems.

Finally, the northern regions provide ideal conditions for solar parks due to high radiation and clear skies. However, central and southern Chile have seen a surge in solar park projects on agricultural land, driven by reduced transmission costs and water scarcity. The Metropolitan region has the highest replacement proportion of *Espinal* and sclerophyllous forests, while the Biobío region has almost entirely converted its photovoltaic park area from agricultural land. This trend highlights the complex relationship between renewable energy, land use, and environmental considerations.

**Author Contributions:** Conceptualization, A.d.P.; writing—original draft preparation, A.d.P. and G.C.-A.; writing—review and editing, A.d.P., G.C.-A. and B.A.-G.; funding acquisition, B.A.-G. All authors have read and agreed to the published version of the manuscript.

**Funding:** This research was funded by the Ministry of Science and Innovation (Next Generation EU) project TED2021-130665B-100, Spain.

**Data Availability Statement:** The original contributions presented in the study are included in the article, further inquiries can be directed to the corresponding author.

**Conflicts of Interest:** The authors declare no conflicts of interest.

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
