# Peer review of "Consequences of Land Use Changes on Native Forest and Agricultural Areas in Central-Southern Chile during the Last Fifty Years"

_land, doi:10.3390/land13050610_

Round 1
Reviewer 1 Report
Comments and Suggestions for Authors
REVIEW REPORT
CODE land-2939104
DATE 27/03/2024 – DEADLINE 29/03/2024
Dear authors,
I think this manuscript is quite interesting but I believe it cannot be published in the present form. I recommend the authors follow the standard structure for study cases (i.e. Introduction, Materials and Methods, Results, Discussion and Conclusions). On the other hand, materials and methods are missing. Authors should clarify which sign (comma or point) indicates decimal part and the separation for thousands. Finally, I think that the “solar photovoltaic park” section does not fit very well in the general discourse; it needs to be integrated with the rest of the text. And you do not need to give a definition of what a solar photovoltaic park is.
Author Response
Thanks for the comments. We reorganize the article using the standard structure of Introduction, Study Site, Materials and Methods, Result and Discussion, and Conclusions. Thus, the manuscript is an “Article”, not a “Review”. We also changed the order of the results for more clarity, starting from i) population and agricultural land, ii) native forest and exotic trees, iii) expansion of urban areas; and iv) solar photovoltaic park. The definition of solar parks was deleted from the text. As a consequence, the reference number and order changed.
We think that the manuscript is now easier to read and understand.
Also, we reviewed the figures, decimal points, and commas for thousands across the manuscript.
Reviewer 2 Report
Comments and Suggestions for Authors
The article titled "Consequences of land use changes on native forest and agricultural areas in central-southern Chile during the last fifteen" draws a parallel between Chile's transformation and transformations in other Mediterranean countries such as Spain and Portugal, highlighting the environmental and economic implications these changes in land use. The text contains interesting numerical summaries. The authors describe research conducted between 1980 and 2020, revealing a sharp increase in the expansion of forest plantations, which is undoubtedly very important.
All text is written in an understandable way, in "acceptable" English.
Remarks: 1. Please pay attention to the readability of Figure 1 b), photos Figure 2.
2. The literature should be corrected in accordance with the journal's guidelines.
3. Why was this area analyzed?
Despite minor criticisms, this article constitutes a valuable contribution and should be accepted for publication. If the author(s) removes the above issues, I recommend publication.
Author Response
Thanks for the positive opinion. We changed the organization of the article; we included the sections of the Study Site, including the area analysed, Materials and Methods, and the order of the Figures. Figures 1 (now Figure 3) and 2 (now Figure 1) were enlarged and improved in quality. The literature was corrected.
Reviewer 3 Report
Comments and Suggestions for Authors
The authors declare in the article: "This study focused on analyzing land use changes in Chile during a fifteen-year period and evaluating their effects on native forest cover, ecosystem services, and agricultural land".
From this statement and the title I expected to read a methodological article. Instead, the article is a collection of data, resembling more a critical review of analyses that have already been done, and it is not very clear how the conclusions are reached. It is true that the article is presented as a review, but the title, abstract and the rest of the text do not show this. It seems to me more like a technical note.
The article however is good, it would only require an effort to better frame the article as a review by stating it from the beginning.
Other minor comments:
- The graphs in figure 1 are very interesting and would require more decription. For example, the graphs in figure 1a and b are never mentioned in the text.
- The diagram in figure 2 should be better explained. What do the arrows of different thicknesses mean? And the dashed ones? Is there a reason as to why they are different?
- L 246 and L 264 figures 2c and 2d are mentioned, which are not present in the text.
Author Response
Thanks for the comments. We reorganize the article using the standard structure of Introduction, Study Site, Materials and Methods, Result and Discussion, and Conclusions, as suggested by reviewer 1. We changed the order of the results for more clarity, starting from i) population and agricultural land, ii) native forest and exotic trees, iii) expansion of urban areas, and iv) solar photovoltaic park, and as a consequence, the order of the Figures were modified. We think that the manuscript is now easier to read and understand. Thus, the manuscript is an “Article”, not a “Review”.
- The graphs in figure 1 are very interesting and would require more decription. For example, the graphs in figure 1a and b are never mentioned in the text.
Thanks for the observation. Figure 1 (now Figure 3) has a better description and has been enlarged for clarity. Also, figures A - D are now mentioned in the text.
- The diagram in figure 2 should be better explained. What do the arrows of different thicknesses mean? And the dashed ones? Is there a reason as to why they are different?
Thanks for the comment. Figure 2 (now Figure 1) was enlarged and all the arrows are now similar.
- L 246 and L 264 figures 2c and 2d are mentioned, which are not present in the text.
Thanks for the observation. That was referred to in Figures 1 C and D; there was a mistake. But now, Figure 1 has changed to Figure 3, and all graphs are mentioned in the text.
Reviewer 4 Report
Comments and Suggestions for Authors
Manuscript ID: land-2939104
Title: Consequences of land use changes on native forest and agricultural areas in central-southern Chile during the last fifteen years
This review paper requires some major revisions. See my concern, specifically in the abstract section
Abstract:
It is not logically to discuss findings with literature review in the abstract section (Referee to line 16 to 17)…. This expansion coincided with a 12.7-27.0% reduction per decade in native forests and a 4.9-23.9% decline in shrublands, including thorn scrub, in central-southern Chile.
Language requires improvement. …Line Urban areas experienced a significant 91% growth in the last 27 years, concentrated in the Mediterranean climate region (line 20).
The abstract lacks methodology. In addition, data sources and method data analysis are not well introduced in the abstract section. Is it meta-analysis or systematic review?
Introduction: novelty and justification should be improved.
Comments on the Quality of English LanguageManuscript ID: land-2939104
Title: Consequences of land use changes on native forest and agricultural areas in central-southern Chile during the last fifteen years
This review paper requires some major revisions. See my concern, specifically in the abstract section
Abstract:
It is not logically to discuss findings with literature review in the abstract section (Referee to line 16 to 17)…. This expansion coincided with a 12.7-27.0% reduction per decade in native forests and a 4.9-23.9% decline in shrublands, including thorn scrub, in central-southern Chile.
Language requires improvement. …Line Urban areas experienced a significant 91% growth in the last 27 years, concentrated in the Mediterranean climate region (line 20).
The abstract lacks methodology. In addition, data sources and method data analysis are not well introduced in the abstract section. Is it meta-analysis or systematic review?
Introduction: novelty and justification should be improved.
Author Response
Thanks for the comments. The abstract was modified completely, and now it includes data sources and method data analysis.
We reorganize the article using the standard structure of Introduction, Study Site, Materials and Methods, Result and Discussion, and Conclusions, as suggested by reviewer 1. So, it is not a meta-analysis or systematic review; it is an “Article”. As indicated now in the Abstract and Materials and Methods, agricultural data for Chile (1980-2020) were obtained from public Chilean institutions (INE and ODEPA). Land use changes in central and south Chile (1975-2018), analysed from satellite images, were obtained from indexed papers. Urban area expansion in Chile between 1993 and 2020 was analysed from MINVIU data.
Novelty and better justification of this work is now more explicit in the Introduction.
Round 2
Reviewer 3 Report
Comments and Suggestions for Authors
The authors modified the paper according to the reviewers' comments. I believe it is suitable for publication.
Reviewer 4 Report
Comments and Suggestions for Authors
Manuscript Title: Consequences of land use changes on native forest and agricultural areas in central-southern Chile during the last fifty years
The authors have addressed all feedback in this paper, ensuring its suitability for publication. Manuscript structure and format have been substantially improved for potential publication. Thus, the current revised version meets the standards of peer review article.